# Spatiotemporal Differences and Spatial Spillovers of China’s Green Manufacturing under Environmental Regulation

**DOI:** 10.3390/ijerph191911970

**Published:** 2022-09-22

**Authors:** Jie Tao, Weidong Cao, Yebing Fang, Yujie Liu, Xueyan Wang, Haipeng Wei

**Affiliations:** School of Geography and Tourism, Anhui Normal University, Wuhu 241002, China

**Keywords:** green manufacturing efficiency, environmental regulation, spatial and temporal divergence, spatial spillover, China

## Abstract

Faced with the real demand of manufacturing industry to achieve the goal of green and high-quality development, exploring spatiotemporal heterogeneity and the spatial spillover effect of green manufacturing efficiency under environmental regulation can help reveal the path and mechanism of green development in the manufacturing industry. By using the SBM-DEM model to measure green manufacturing efficiency at the urban scale in China, exploratory spatial analysis is used to characterize the spatiotemporal differentiation of urban green manufacturing efficiency from 2003 to 2018. With the help of the spatial Durbin model, the impact of environmental regulation on green manufacturing efficiency and the spatial spillover effect are demonstrated. The results show that: (1) The green manufacturing efficiency of cities has developed in a gradual and balanced manner in time series, and the degree of equalization is stronger in the eastern coast than in the western inland; (2) Urban green manufacturing efficiency patterns are misaligned with economic scale patterns, indicating that green manufacturing is not traditionally dominated by economic factor inputs; (3) The practice of Chinese cities has proved that environmental regulation can significantly inhibit the development of green manufacturing efficiency in local cities. The crowding-out effect and optimization effect of environmental regulation on other external factors indirectly affect green development. By comparing different spatial weight matrices, it is shown that the economic relationship between cities can offset the inhibition of environmental regulation; (4) Although environmental regulation under spatial interaction would have significantly contributed to the green manufacturing efficiency of neighboring cities, this contribution effect is insignificant and weak due to the economic interactions between cities. Empirical research provides a theoretical foundation for the development of green manufacturing from the standpoint of environmental regulation, allowing green development research in manufacturing to move further.

## 1. Introduction

The establishment of an ecological civilization is an important part of the Chinese government’s new development model. Therefore, the Chinese government’s economic development strategy has shifted from an extensive pursuit of scale growth to a connotative development with “ecological priority and green development” as the basic guide to improve economic quality and efficiency. Manufacturing is an important vehicle for China to achieve green economic development and high-quality development [1,2]. In the global competition landscape, its scale level has reached the world’s first level for many consecutive years [3]. However, the contradiction between the reality of improving manufacturing efficiency and non-desired high pollution output of manufacturing resource inputs is highlighted. Environmental regulation in China is considered an important means of solving environmental and resource problems, and the manufacturing industry, as the foundation of the country, bears a special burden and historical responsibility for green development [4]. Therefore, to meet the realistic needs of green and high-quality development of the manufacturing industry, it is an essential step to interpret the spatial-temporal differentiation and the spatial spillover effect of green manufacturing efficiency under environmental regulation [5]. At the same time, this is conducive to improving the theory of green development in the manufacturing industry and exploring the path of green development.

Research on explaining China’s economic growth miracle, in terms of total factor productivity in manufacturing, based on an input-output perspective, have a long-standing since the 1990s, since which time the sustainability of the Chinese economy has been called the “Southeast Asian miracle” [6,7]. It is argued that China’s manufacturing sector cannot achieve sustainable development by relying solely on the crude growth of external factors and that there are doubts about the sustainability of total factor productivity gains [8]. Scholars, represented by Brandt et al. [9,10], analyzed the interaction between tariff liberalization and the entry and exit of firms from the market and concluded that total factor productivity in China has achieved effective growth. Early studies tended to examine the macroeconomic effects from the perspective of analyzing the development and time-series evolutionary characteristics of the total factor productivity in manufacturing in the whole country or region, using capital, labor and other factors as the main inputs emphasized by classical economics [11,12]. In the pursuance of green development, the overall characteristics of the manufacturing industry are measured by data envelopment and other analysis methods, with environmental pollution limiting variables as non-desired outputs [13,14]. As the foundation of this, studies related to spatiotemporal heterogeneity and industry heterogeneity have been conducted, which have become an important research element for the current green development of the manufacturing industry [15,16,17]. Meanwhile, the implementation of environmental governance measures by the government, as the main means of economic green development, has become a research hotspot [18,19].

Scholars mainly concentrate on the research of environmental regulation measurement and environmental regulation effects. They use a single indicator to measure pollution discharge, environmental cost or environmental investment [20,21]. With the deepening of research, several indicators have been used, such as environmental cost and environmental investment, to evaluate the explicit environmental regulation of the government and the integrated indicator method has been further used to construct the intensity of environmental regulation, based on the actual pollution indicators of each industry [22,23,24]. There are existing studies on the effects of environmental regulation that generally agree that the roles of firms’ technological innovation, industrial economic performance and foreign direct investment are important [25,26,27]. Judging from the impact of environmental regulation on economic performance, some scholars believe that environmental regulation increases the operating costs of enterprises, thus, reducing the productivity of enterprises [28]. Additionally, environmental regulations are beneficial to technological innovation to a certain extent, and this influence is greater than the negative influence of environmental regulation on cost increase, thus, prompting enterprises to continuously improve productivity and, finally, realize an improvement of economic performance [29]. With the deepening of the interdisciplinary theoretical research path, the mechanism of environmental regulation, as a powerful means of green economic development, has gradually become a hot topic [30,31,32].

Existing research has fully demonstrated the economic externalities and environmental externalities of green development of the manufacturing industry and achieved full research results, but there are still some issues that need to be deeply considered, such as the following: (1) Exploring the growth mode and regional differences of manufacturing efficiency, with economics and management as the main research disciplines, focusing on exploring the relationship between quantitative change characteristics and mathematical measurement, ignoring the “pattern–process-mechanism” research on manufacturing efficiency emphasized by geography; (2) Against the background of China’s high-quality economic development, the model for measuring manufacturing efficiency from the perspective of input-output should consider the unexpected output of environmental pollution. It is untoward to scientifically evaluate the actual level of green development of manufacturing industry by constructing a manufacturing efficiency measurement model only from the perspective of capital, labor and other factors; (3) The existing studies are insufficient in describing the spatial and temporal evolution of manufacturing efficiency by city-scale environmental regulation. A finer scale can more accurately express the spatial and temporal heterogeneity of manufacturing green development.

## 2. Theoretical Foundation

To achieve and support the goal of global sustainable development, the proposal of green development theory has profound theoretical connotations and practical pointers for promoting sustainable development in countries around the world [33]. Therefore, the manufacturing industry, to achieve green development, needs to incorporate ecological and environmental constraints into its economic development framework, with ecological values and ecological ethics as the leading concepts to achieve “greening” of manufacturing technology upgrading, industrial adjustment, and management methods [34,35].

As an important practical tool for governments to achieve green economic development, academics have long conducted theoretical investigations on the topic of environmental regulation for economic development efficiency. On the one hand, the impact of environmental regulation on efficiency change is mainly realized by changing allocative efficiency and scale efficiency [36]. The government’s main purpose of environmental regulation is to impose constraints on the environmental pollution behavior of enterprises and to make environmental resources enter into the production decisions of enterprises as an input factor, such as capital and labor, by assigning a price to environmental resources. Environmental regulation actually opens up the resource allocation channel through the improvement of the resource price mechanism and resource management mechanism, such that the flow of resources follows the price signal, and the improvement of resource allocation efficiency under the given input scale significantly improves production efficiency [37,38,39]. On the other hand, according to the theory of endogenous growth, the progress of productivity strongly relies on the growth rate of technological progress [40]. The influence of environmental regulation on technological innovation progress is derived from Porter’s hypothesis, the core of which is that environmental regulation can effectively stimulate the innovative behavior of enterprises in technology [41], resulting in an “innovation compensation effect”, including products, equipment and production technology, which can compensate for, or even exceed, the compliance cost [42].

From the perspective of spatial differences in manufacturing enterprises, under the dual pressure of environmental regulation and industrial transformation and development, less developed regions receive heavily polluting enterprises from developed regions and become the “pollution refuge” of developed regions [43,44]. Furthermore, the “polarization-trickle-down” effect theory holds that the relatively underdeveloped areas of the manufacturing industry acquire advanced technology and management experience from the economically developed areas of the manufacturing industry through technology introduction, the knowledge spillover effect or the “learning effect”, and the developed areas of the manufacturing industry constantly promote the gathering of favorable factors, such as talent and capital, thus, promoting the economic heterogeneity and spatial heterogeneity of manufacturing industry development [45]. On the basis of these studies, the theoretical framework of manufacturing industry’s spatial-temporal differentiation and spatial spillover effect under environmental regulation is constructed in this paper (Figure 1).

## 3. Methods and Data

### 3.1. Urban Green Manufacturing Efficiency

#### 3.1.1. Indicator System Construction

The evaluation of green manufacturing efficiency at the urban scale is a necessary and highly objective basis for comprehensively considering environmental impact and resource efficiency modernization, and the green total factor productivity in manufacturing, measured from input-output perspective data. As an important bearer of China’s industrial modernization development, secondary industry data are used instead of manufacturing industry data in this paper, due to the fact that availability of manufacturing industry data makes a small difference between manufacturing industry statistics and secondary industry statistics. To measure the efficiency of China’s green manufacturing, it is necessary not only to scientifically and reasonably measure manufacturing efficiency from the input–output perspective, but also to fully consider resource consumption and environmental pollution. Therefore, a green manufacturing total factor productivity index system, including unexpected output, is constructed. In the index system, the amount of urban fixed asset input, industrial electricity consumption, the number of industrial employees and the built-up area represent the input factors of the manufacturing industry. The actual GDP is regarded as the output parameter, and the emissions of industrial wastewater, sulfur dioxide, and soot are used to replace the non-desired output parameter of environmental pollution in the manufacturing industry. The fixed asset input is based on the idea of Young [46], so the perpetual inventory method is used to obtain the stock of fixed assets in all years, and the actual GDP is deflated by the nominal GDP of each year, according to the price index with 2003 as the base period. In this paper, the original data were obtained from the China Regional Statistical Yearbook and the City of China Statistical Yearbook. Due to the large total sample size, the missing values of panel data were completed by linear interpolation.

#### 3.1.2. Calculation Method

To incorporate environmental pollution variables into the economic efficiency calculation framework, the super-efficient SBM-DEA, considering non-desired outputs, was proposed, by improving the model based on the SBM-DEA model proposed by Tone [47]. The super-efficiency model could break through the bottleneck limit with an efficiency value of 1 and further decompose the effective decision-making units, which could ensure the model results’ accuracy. Its evaluation model is:(1)$$min\rho^{*}=\frac{1+\frac{\frac{1}{m}\sum_{m=1}^{M}s_{m}^{x}}{x_{jm}^{t}}}{1-\frac{1}{l+h}\left(\frac{\sum_{l=1}^{L}s_{l}^{y}}{y_{jl}^{t}}+\frac{\sum_{h=1}^{H}s_{h}^{b}}{b_{jh}^{t}}\right)},\{\begin{array}{l} x_{jm}^{t}⩾\sum_{j=1,j\neq 0}^{n}\lambda_{j}^{t}x_{jm}^{t}+s_{m}^{x}\\ y_{jl}^{t}⩾\sum_{j=1,j\neq k}^{n}\lambda_{j}^{t}y_{jl}^{t}-s_{l}^{y}\\ b_{jh}^{t}⩾\sum_{j=1,j\neq k}^{n}\lambda_{j}^{t}b_{jh}^{t}+s_{h}^{b}\\ \lambda_{j}^{t}⩾0,s_{m}^{x}⩾0,s_{l}^{y}⩾0,j=1,\cdots ,n \end{array}\;.\;$$
where $\rho^{*}$ is the green manufacturing efficiency value; $x_{j}^{t}$ denotes the input parameters of j in period *t*; $y_{j}^{t}$ and $b_{j}^{t}$ are the desired output parameters and non-desired output parameters, respectively; m, l and h denote the number of corresponding factors; $s_{m}^{x}$ and $s_{l}^{y},\;s_{h}^{b}$ denote the corresponding slack vectors, respectively; and λ is the decision unit weight vector.

### 3.2. Kernel Density Estimation

Kernel density estimation is a nonparametric method for estimating probability functions. The smooth peak function was used to fit the data points of the green manufacturing efficiency of Chinese cities in each year, and, finally, the distribution probability distribution curve was simulated. The formula is:(2)$$f\left(x\right)=\frac{1}{nh}\sum_{i=1}^{n}K_{0}\left(\frac{\left|x-x_{i}\right|}{h}\right)$$
where f(x) is the kernel density function expression; $K_{0}x$ is the kernel function parameter; n is the sample capacity parameter; h > 0 is the smoothing index, also known as bandwidth or window; x is the overall green manufacturing efficiency parameter; and $x_{i}$ represents the green manufacturing efficiency parameter of city i.

### 3.3. Hot Spot Analysis

Hotspot analysis is a typical method to explore spatial clustering, with the help of the index to discern the clustering of high and low values of green manufacturing efficiency in Chinese cities. The calculation formula is:(3)$$G_{i}^{*}=\frac{\sum_{i=1}^{n}w_{i,j}\left(d\right)x_{i}}{\sum_{i=1}^{n}x_{i}}\;$$

To facilitate comparative analysis, the *z* value of $\mathrm{Getis-Ords}\;G_{i}^{*}$ is standardized:(4)$$Z\left(G_{i}^{*}\right)=\frac{G_{i}^{*}-E\left(G_{i}^{*}\right)}{\sqrt{Var\left(G_{i}^{*}\right)}}$$
where $E\left(G_{i}^{*}\right)$ and $\sqrt{Var\left(G_{i}^{*}\right)}$ are the expectation and variation values of $G_{i}^{*}$, respectively; $w_{i,j}$ is the spatial weight; and $x_{i}$ is the green manufacturing efficiency of city i. If $Z\left(G_{i}^{*}\right)$ passes the significance test and $Z\left(G_{i}^{*}\right)>0$, it is a hot spot with high green manufacturing efficiency; otherwise, it is a cold spot.

### 3.4. Space Panel Durbin Model

The spatial panel Durbin model is mostly used to explore the spatial relationship and interaction of geographical things. This paper studied the spatial dependence effect between independent variables, such as environmental regulation and the dependent variable green manufacturing efficiency. Selecting the appropriate spatial panel model was the key step of parameter estimation. It was determined to be the spatial Durbin model through inspection and reconstruction in this paper, so it is not described again.

The selection and construction of spatial weights are the key steps of spatial econometrics and the basis for the rational operation of spatial econometric models. In view of this, there is still no accurate conclusion about matrix selection in academic circles, and the following three spatial weight matrices were selected:(1)Geographic distance matrix ($W_{d}$). The first law of geography in geography states that spatial influence presents as a characteristic of decay with spatial distance [48], so the inverse of the Euclidean distance between cities within the study sample was used to construct the weight matrix, and the formula is:
(5)$$W_{\lambda \gamma }=\{\begin{array}{l} 1/S_{\lambda \gamma },\;\lambda \neq \gamma \\ 0,\;\;\lambda =\gamma \end{array}$$(2)Economic distance matrix ($W_{d}$). The close economic ties of spatial units with different economic development levels can reflect their actual economic spatial relationships through distance, so the economic distance matrix with the difference in GDP per capita among research units as the measurement index was constructed, and the formula is:(6)$$W_{\lambda \gamma }=\{\begin{array}{l} 1/\left|X_{\lambda }-X_{\gamma }\right|,\;\lambda \neq \gamma \\ 0,\;\;\;\;\;\;\lambda =\gamma \end{array}$$
where $X_{\lambda },$ $X_{\gamma }$ represent the per capita GDP of cities λ, γ respectively.(3)Economic-geographic distance nested matrix ($W_{ed}$). In fact, the economic development and closeness of linkage of spatial units is the result of the joint action of multiple factors, such as economic, cultural and institutional factors, and measuring the degree of spatial linkage simply by spatial distance or economic distance triggers certain errors. Therefore, the nested matrix of economic-geographical distance was used to measure the difference in regional unit spatial connections, and the formula is:(7)$$\begin{array}{l} W_{\lambda \gamma }=W_{d}\times diag\left(\frac{{\bar{Y}}_{1}}{\bar{Y}},\frac{{\bar{Y}}_{2}}{\bar{Y}},\cdots ,\frac{{\bar{Y}}_{n}}{\bar{Y}}\right)\\ {\bar{Y}}_{\lambda }=\frac{1}{t_{1}-t_{0}+1}\sum_{t=t_{0}}^{t_{1}}Y_{\lambda t},\bar{\;Y}=\frac{1}{n\left(t_{1}-t_{0}+1\right)}\sum_{i=t_{0}}^{t_{1}}\sum_{i=1}^{n}Y_{\lambda t} \end{array}$$
where $W_{\lambda \gamma }$ is the nested spatial weight of the economic-geographic distance between cities λ and γ; $W_{d}$ is the constructed spatial weight matrix of geographic distance; ${\bar{Y}}_{\lambda }$ is the average actual GDP of city λ from 2003 to 2018; and $\bar{Y}$ represents the sum of the actual GDP of all cities from 2003 to 2018.

The data content of this paper is described as follows (Table 1).

## 4. Analysis of the Results

### 4.1. City Green Manufacturing Efficiency Measurement

#### 4.1.1. Changes in Regional Characteristics

From 2003 to 2018 (Figure 2), the trend of the green manufacturing kernel density function curves in Chinese cities remained stable, their peak inflection points were all below 0.5 of green manufacturing efficiency, and the kernel density values of cities with green manufacturing efficiency higher than 1 continued to increase, indicating that green manufacturing in Chinese cities generally changed from inefficient crude production to efficient production. The wave height of the kernel density curve decreased continuously with time, which indicated that China’s urban green manufacturing efficiency had a balanced development trend and that China’s resource-saving and environmentally friendly progressive development model promoted the development of urban green manufacturing efficiency. However, in 2018, compared with 2013, there was a decline, which highlighted that the Chinese government attached great importance to ecological environment construction, and it also showed that strong environmental regulations inhibited the green production efficiency of Chinese urban manufacturing industry.

Furthermore, to reflect regional differences, the cities in the study sample were divided into four regions belonging to the east, west, central, and northeast [49]. In four research years (Figure 3), the inflection points of the regional nuclear density peak was lower than 0.5, which was in alignment with the trend toward green manufacturing efficiency in China. By region, green manufacturing efficiency had a trend of equalization, with the eastern region ranking first, followed by the central, northeastern and western regions, in that order. The green manufacturing efficiency of the eastern cities showed an explosive development stage of efficient cities, and the gap between the number of green manufacturing efficiency cities and other regions widened. The green manufacturing efficiency peak inflection point in central cities kept the trend of a rising peak. The Northeast gradually overtook the central region from the low green manufacturing efficiency cluster. In 2013, the nuclear density curves of the two remained basically the same, and there was a great tendency to catch up. However, the peak of the kernel density curve in Northeast China in 2018 was significantly lower than that in Central China, indicating that the efficiency of green manufacturing in the old industrial base in Northeast China, where heavy industry is the main production industry, was more affected by environmental regulations. Regarding the green manufacturing efficiency of western cities, although it was the region with the lowest peak value of nuclear density, it was the region with the greatest development potential. Compared with the nuclear density curves in central and northeast China in 2003, the difference of nuclear density curves in western China in 2018 was significantly reduced, indicating that the difference in green manufacturing efficiency among regions was shifting to balanced development.

#### 4.1.2. Spatial-Temporal Agglomeration and Differentiation

Considering the similarity of the structures of green manufacturing efficiency in Chinese cities studied, and the limitation of article length, four cross-sectional datasets from 2003, 2008, 2013, and 2018 were selected for cold hotspot analysis, which showed the spatial and temporal variation characteristics of green manufacturing efficiency in Chinese cities and facilitated the drawing of a cold hotspot clustering map, based on significance, to reveal the spatial pattern of green efficiency in Chinese cities (Figure 4). In terms of spatial distribution, the pattern of urban green manufacturing efficiency is misaligned with the pattern of the urban economic scale, such as Yangtze River Delta and other top economic scale urban clusters, which are not very efficient in green manufacturing under environmental regulation, although they had vast economic and material factor inputs. This suggests that green manufacturing efficiency is no longer dominated by economic inputs in the traditional sense, while environmental regulations could limit the development of green manufacturing efficiency.

China’s green manufacturing efficiency has a strong “polarization” phenomenon. In terms of spatial agglomeration, high green manufacturing efficiency is mainly concentrated in developed cities along the southeast coast, while low green manufacturing efficiency is mainly concentrated in the urban spatial fringes of the sample. The main reasons for this spatial change can be attributed to two aspects. First, compared with inland cities, coastal cities have significant location advantages, and even higher openness to the outside world is more propitious to attracting the influx of advanced technology, talents, capital, and other manufacturing basic elements from outside the region, further stimulating the effective promotion of green manufacturing efficiency and spatial agglomeration [50]. Second, in the world industrial division chain, manufacturing production is the main link undertaken by China, and the high demand for natural resources and labor is an inherent attribute of manufacturing production. The main agglomeration area of high green manufacturing efficiency is in the densely populated area of the Hu Huanyong line, and the sufficient labor force and resource endowment can provide sufficient guarantees for the development of labor-intensive light processing and high energy-consuming enterprises in large- and medium-sized cities in coastal areas [51]. The differentiation of manufacturing endowment among cities eventually forms cold hotspot agglomeration divergence.

In terms of temporal evolution, as China’s environmental regulations continue to strengthen green manufacturing efficiency, hotspots along the southeast coast are gradually transformed into sub-hotspots. Some cities in Northeast China changed from insignificant agglomeration to hotspot and sub-hotspot agglomeration and, finally, evolved to insignificant agglomeration, indicating that Northeast China, as an old industrial base dominated by heavy polluting industries, shows insignificant agglomeration, due to the constraints of resources and environmental regulations while improving manufacturing efficiency to meet the needs of national economic development. The cities in the southwest changed from hot spots to cold spots, and the adjacent Chengdu-Chongqing urban agglomeration gradually formed the Sichuan-Shandong hot spot agglomeration belt connected with Hubei Province, Henan Province, and Shandong Peninsula.

On the whole, the coastal high green manufacturing efficiency agglomeration belt and the Sichuan-Shandong high green manufacturing efficiency agglomeration belt will eventually form, and the cold spot agglomeration belt, with clear boundaries, will form beside the two belts. This shows that hot city agglomeration has a siphoning effect on production factors, such as capital and talent in neighboring cities, which eventually promotes the formation of an evolutionary divergent pattern of green manufacturing efficiency agglomeration in China. According to the traditional logic of economic inputs driving manufacturing efficiency, the green manufacturing efficiency of Chinese cities does not show a clear Matthew effect, which also indicates that green manufacturing efficiency is simultaneously constrained by factors such as environmental regulations.

### 4.2. Empirical Study on Environmental Regulation Effect

#### 4.2.1. Variable Setting

(1)Explained variable.

From the factor input–output perspective, Green Manufacturing Efficiency (GME), an efficiency level surveyed by SBM-DEA model, was used as the explained variable.

(2)Core explanatory variable.

The “cost of compliance” effect states that a certain intensity of environmental regulation can prompt enterprises to incur increased explicit cost in dealing with pollution and implicit costs of communicating with the government; thereby reducing the production efficiency. In contrast, Porter proposed that moderate pressure from environmental regulation on the production activities can effectively promote the technological innovation of enterprises and enhance their competitive advantages in production activities [37]. Under China’s increasingly severe environmental protection environment, government environmental regulation has long formed a green barrier for the development of urban manufacturing. Therefore, in this study, the environmental regulation index (ER) was measured using industrial emissions of three wastes as the raw data, drawing on the approach of Ye et al. [52].

(3)Control variables.
Industrial Structure (IS). Due to the manufacturing industry belonging to secondary industry, the differentiated green development of industries has heterogeneity in economic output, environmental output and green efficiency [53]. To avoid multicollinearity among influencing factors, the ratio of the value added of secondary and tertiary industries, which reflects industrial upgrading, was applied to characterize the change in industrial structure.Economic development level (EDL). Taking the “Environmental Kuznets Curve” as the mainstream hypothesis, this paper expounded the relationship between economic growth and environmental development. So, economic growth was an important indicator that affected urban development, and economic development inevitably affects urban green manufacturing efficiency. Therefore, GDP per person was selected as the quantitative index [54].Innovation capability (IC). Digitalization and intelligence are the common trends of current manufacturing development, and innovation drive is an important push to achieve the improvement of manufacturing technology and management technology [34]. To measure the innovation capacity of cities in innovation activities, the proportion of local science and technology expenditure was considered as its proxy.Government Intervention Degree (GID). The inhibitory effect of “government failure” on urban green development has been confirmed by scholars, while some scholars believe that appropriate government intervention can effectively promote the urban innovation environment and infrastructure construction to provide basic guarantees for green all-factor improvement [55,56,57]. Fiscal expenditure is an important indicator of the degree of government intervention, so the share of fiscal expenditure was used as a token variable.Degree of external openness (DOEO). Foreign direct investment brings the externalities of a “pollution paradise” or “pollution halo”, which leads to a change in the green manufacturing efficiency of the city. Therefore, we chose the proportion of foreign-invested enterprises as the representation.(4)Variables set pre-test

① Correlation analysis.

The correlation degree among variables was verified, and the results show that all variables, except the variables controlled by the degree of external openness, passed the significance test, so all the selected variables, except LnDOEO, were suitable for building the model (Figure 5).

② Collinearity diagnosis.

The variables that passed the initial correlation test were brought into the backward stepwise regression, and the model results showed that the variables’ VIFs were all below 5, so they all passed the multicollinearity test (Table 2). According to the empirical measurement experience, all the variables, except the LnDOEO control variables, were suitable for building the space panel model.

#### 4.2.2. Model Selection Test and Model Reconstruction

To use the spatial panel model accurately and to specify the specific form, the test steps were as follows. First, we specified whether there was spatial correlation in the data under different weight matrices, which was reflected by Moran’s I. Then, the original hypotheses of both the LM and LR tests were whether the spatial panel model could be degraded to other spatial panels, such as spatial lag models or spatial error models. The null hypothesis of the next Hausman test was that the model applied to random effects, and the explicit model choice was random effects or fixed effects. Finally, we borrowed the joint effects test to clarify whether the time effect applied. The spatial correlation test of significance under different weight matrices, using Moran’s I, proved to be a pass, so the spatial panel model could be used (Table 3). The results showed that the spatial correlations of the different matrix models passed the test and all significantly rejected the null hypothesis of LM. On this basis, further LR tests were performed, and the results showed that the different spatial weight matrix models rejected the null hypothesis that the selected SDM model could not degenerate into other spatial panel models. The Hausman test results showed negative statistics under the *W_d_* and *W_ed_* spatial weight matrices, while rejecting the null hypothesis under the other matrices, so the fixed model was chosen for both [58]. Finally, the results of the borrowed joint effects test showed the use of a temporal and spatial dual fixed space Durbin model. Consequently, the reconstruction model was:(8)$$\ln GME_{i,t}=\mu_{i}+\rho W\ln GME_{i,t}+\alpha_{1GME}\ln ER_{it}+\alpha_{2GME}X_{it}+\lambda_{1GME}W\ln ER_{it}+\lambda_{2GME}WX_{it}+\varepsilon_{it}$$
where W represents the constructed spatial weight matrix, and this study used the spatial distance $W_{d}$ of order 284 × 284, and the economic distance and the nested matrix $W_{ed}$ considering the economic and geographic spatial distances. The estimated coefficient ρ of the spatial regression characterizes the estimated value of the explanatory variable $\ln GME_{i,t}$,$\;\alpha_{1GME}$ and $\alpha_{2GME}$ explain the estimated values of regression coefficients of variables and $\lambda_{1GME}$ and $\lambda_{2GME}$ are the spatial regression coefficients of explanatory variables. $X_{it}$ is the control variable; $\mu_{i}$ is the individual fixed effect value not observed by the model; $\varepsilon_{it}$ is the random error term.

#### 4.2.3. Environmental Regulation Parameter Estimation Analysis

(1) Overall effect estimation.

The model estimates were constructed based on different spatial weight matrices using Stata showing generally consistent results, proving that the model parameter estimates were basically robust (Table 4). In terms of environmental regulations affecting urban green manufacturing efficiency, the LnER coefficients passed the 0.001 significance level at different spatial weight matrices with coefficient values of −0.0448, −0.0427, and −0.0415, indicating that environmental regulations had a significant inhibitory effect on the efficiency of urban green manufacturing. The coefficient of estimated spatial lag term Wx * LnER was 0.0877, and it passed the significance test of 0.001 in the estimation of geographical distance spatial matrix model. This meant that environmental regulation could significantly promote green manufacturing efficiency in the surrounding areas, only considering the spatial correlation of geographical distance. The coefficients of Wx * LnER under the economic distance spatial matrix and economic-geographic distance spatial matrix were 0.0078 and 0.0125, respectively, which were positive but did not pass the significance test, indicating that the spatial spillover effect of environmental regulations on green manufacturing efficiency in the surrounding areas was not significant. The non-significance of the spatial spillover benefits in the empirical evidence when considering the spatial correlation of the study region’s economy suggested that economic linkages between cities and neighboring cities could offset the positive spatial spillover effects of environmental regulations to some extent.

According to the industrial gradient theory, industrial restructuring in developed cities, for the purpose of enhancing ecological and environmental quality development, shift heavy pollution manufacturing industries to neighboring cities that are relatively less developed, while developed cities have comparative advantages in economic activities, which are more attractive to capital, human and other factors in less developed cities, and the lack of comparative advantages of cities causes the spatial effect of environmental regulation to be non-significant [59]. Among them, the control variables LnIS and LnGID coefficients were positive, indicating that IS and GID could effectively promote the green manufacturing efficiency of local cities, while the LnEDI and LnIC coefficients were negative, indicating that EDI and IC inhibited the development of green manufacturing efficiency to some extent, and the spatial lag estimation coefficient of control variables failed the significance test, so their spatial spillover effect was not obvious. Since Lesage pointed out the possibility of estimation error for the overall point estimates of the parameters [60], which can only be used as a preliminary analysis of spatial spillover effects, further partial differential estimation decomposition of the point parameter estimates was needed.

(2) Total effect decomposition.

To further study the specific effect of environmental regulation on green manufacturing efficiency, the partial differential estimation decomposition of the total effect and the estimation results of its effect decomposition parameters under different spatial weight matrices are shown in the following Table 5.

Initially, this research part analyzed the impact of environmental regulations on local green manufacturing efficiency. The estimated coefficients of the direct effect of environmental regulation on local green manufacturing efficiency were −0.0444, −0.0425, and −0.0414 under the significance level test of 0.001 in different models, certificating that green manufacturing efficiency in local areas was significantly inhibited by environmental regulations. There were some possible reasons for this result. First, China’s manufacturing development is in line with the “cost of compliance” hypothesis, in which the intensity of environmental regulations in the current prefecture-level administrative units of China continues to increase the environmental compliance costs of the manufacturing [61]. This has led the manufacturing industry to shift from rampant outward development to high-quality inward development, which has forced traditional manufacturing processes to change and has led manufacturing companies to give weight to innovation development, and intelligent digitalization development of the manufacturing, and the increase in research activities have brought a cost burden [62]. Second, industrial upgrading and industrial transformation of manufacturing industry, which aims at achieving green development, leads to the loss of comparative advantage of manufacturing product prices in the opening-up competition pattern at the initial stage of transformation and upgrading. Factor-oriented cost pressures force manufacturing industries to restructure their industries to achieve a reduction in local undesired output and improve local green manufacturing efficiency in terms of quality [63]. Third, local governments are the main implementers and policy-makers of environmental regulations. Although government industrial development policies can effectively promote the efficiency of manufacturing industries, manufacturing industries have to increase the explicit costs of equipment and technology, combined with experience in dealing with pollution, and increase the implicit costs of communication with the government to cope with the actual needs of environmental regulations.

By comparing the increasing phenomenon of the estimated coefficients of environmental regulations in different weight matrices, it was shown that intercity economic correlation could reduce the inhibitory effect of environmental regulations. This is because economic linkages between different cities can promote the mobility of technology, experience, policies, and other elements to achieve environmental regulations, and the economic radiation from neighboring cities economically compensates for the inhibitory effects of local environmental regulations, thus offsetting the negative effects of environmental regulations to some extent. However, ultimately, environmental regulations could still significantly inhibit local green manufacturing efficiency.

Subsequently, the spatial spillover effect was emphatically analyzed. The estimated coefficients of indirect effects of environmental regulations were positive and decreasing under different spatial weight matrices, and their coefficients passed the significance test when only spatial geographical distance was considered, while the estimated coefficients of indirect effects did not pass the significance test when spatial distance matrices with economic factors were considered. The results indicated that the spatial spillover effects were more influenced by geospatial distance, and environmental regulations showed insignificant promotion effects on manufacturing productivity in surrounding areas when economic linkages were considered. This might be attributed to the following: to begin with, after local environmental regulation presented a suppressive effect on local green manufacturing efficiency, the reduction of local undesired environmental pollution output indirectly enhanced the ecological environment of the surrounding cities and reduced the pressure on the manufacturing industry in the surrounding area, thus, promoting green manufacturing efficiency in the surrounding area. In addition, the technology and experience of environmental regulations spilt over to the neighboring regions through geographical and economic connections, indirectly cutting the hidden costs of environmental regulations in the neighboring regions for achieving green development [64], reducing the burden for the manufacturing industries in the neighboring regions and indirectly promoting the green production efficiency of their manufacturing industries. Furthermore, due to the enhancement of urban economic ties, the economic ties compressed the connection distance between cities, making cities with different comparative advantages more able to attract capital, labor and other basic factors of development from neighboring cities [65]. The siphon effect drove the green manufacturing efficiency of neighboring regions to be relatively lower, and the spatial spillover effect was, thus, correspondingly weakened.

### 4.3. Theoretical Framework Construction of Manufacturing Green Development from the Perspective of Environmental Regulation

This paper further constructed the theoretical framework of green development of manufacturing industry through the empirical analysis of the space-time difference and spatial spillover effect of green manufacturing efficiency in China under environmental regulation. The mechanism of environmental regulation on the green development of the local manufacturing industry was interpreted, and its spatial spillover effect on the green development of neighboring manufacturing was expounded in this framework (Figure 6).

For one thing, environmental regulations increased the explicit and implicit costs of green development in manufacturing, thus, directly affecting the efficiency of local green development. At the same time, environmental regulations aimed at achieving green development could produce a pushback effect and optimization effect, which, in turn, led to manufacturing industry restructuring, industrial transformation and upgrading, technological reform and innovation, industrial transfer optimization, ecological and environmental protection enhancement, government scientific intervention and other specific paths indirectly affecting the green development of the local manufacturing. Thus, the local effect was formed by the superposition of the direct and indirect effects of environmental regulation. For another thing, environmental regulations under the role of regional geographical distance and economic linkages had their own spatial spillover effects, and the inhibitory effect of environmental regulations, ecological environment optimization, and the diffusion of technological experience formed positive spillover effects in the surrounding areas. Through empirical construction, the framework provided a theoretical basis for green manufacturing development from the perspective of environmental regulation and opened up a possible realization path for its study.

## 5. Conclusions

### 5.1. Distribution of Cluster Centers

In this paper, from the input–output perspective, the environmental constraint variables were considered non-desired outputs, which were then incorporated into the superefficient SBM-DEM measurement model to measure the green manufacturing efficiency of 284 prefecture-level and above cities in China. This study then explored the spatial heterogeneity of green manufacturing efficiency in Chinese cities by combining exploratory spatial analysis, clarified the spatial spillover effects of green manufacturing efficiency in China under environmental regulation with the help of a spatial panel model, and revealed the theoretical framework of green manufacturing development from the perspective of environmental regulation. The main findings are as follows:

(1) China’s urban green manufacturing efficiency has a progressive and balanced development trend, demonstrating that China’s resource-saving and environmentally friendly progressive development model promotes the development of urban green manufacturing efficiency. Meanwhile, the course of the paradoxical decline in 2013, compared to 2018, could initially reflect the Chinese government’s emphasis on environmental regulations inhibiting the green development of manufacturing. The green manufacturing efficiency equalization degree is strongest in the east, followed by the central, northeast and west, and the overall green manufacturing efficiency equalization degree shows the characteristics of being stronger from the coast than inland.

(2) The pattern of urban green manufacturing efficiency is misaligned with the pattern of urban economic scale, and the green manufacturing efficiency of cities with high factor mobility is not high, indicating that green manufacturing efficiency is no longer dominated by economic inputs in the traditional sense. In the meantime, green manufacturing efficiency has a strong “polarization” phenomenon, with high green manufacturing efficiency mainly concentrated in developed cities on the southeastern coast and low green manufacturing efficiency mainly concentrated in the spatial periphery of China.

(3) China’s urban practice proves that environmental regulation can significantly inhibit the development of green manufacturing efficiency in local cities. In different spatial weight matrices, the green manufacturing efficiency in local cities decreased by 0.0444%, 0.0425% and 0.0414% for every 1% increase in environmental regulation intensity. By comparing the increasing phenomenon of the estimated coefficients of environmental regulations in different weight matrices, it was illustrated that intercity economic correlation could reduce the inhibitory effect that originated from environmental regulations. Although the spatial spillover of environmental regulation significantly promotes the green manufacturing efficiency in neighboring cities under spatial interaction, it shows a nonsignificant and weak promotion effect by economic interaction between cities.

### 5.2. Discussion

On the basis of this research, combined with the shortcomings of this paper, we think it can promote the following further in-depth research:

(1) The comparative analysis of the linear and nonlinear relationships between the effects of environmental regulations on green manufacturing efficiency needs to be strengthened. This study explored the influence of environmental regulation on green manufacturing efficiency in China strictly from a linear relationship, with the help of a spatial panel model, while the green development of the manufacturing industry under environmental regulation is itself a complex dynamic change process under the interaction of multiple factors, and environmental regulation for the purpose of green development often has a nonlinear relationship with the mediating effect on the green manufacturing efficiency of the manufacturing industry. Thus, it is necessary to explore the nonlinear relationship between environmental regulation and green manufacturing efficiency with the help of nonlinear models, such as panel threshold regression, generalized summation, or machine learning-based random forest and XG-Boost models, using factors such as unevenness of urban resource endowment, unevenness of economic development, unevenness of cultural activities and even policy regulation at different development stages as transformation variables.

(2) The study of spatial and temporal differences and multiscale spatial heterogeneity of the green manufacturing industry under environmental regulation has yet to be enriched. The sample of this study was 284 cities at the prefecture level and above in China. We found that a relatively small amount of data in some sample cities was discrete, which might have led to errors in model estimation. Subsequent studies can clean and integrate the data. While manufacturing industry segmentation research has not been considered, a longitudinal study could be conducted on the green development of manufacturing industries in subsectors to summarize their green development industry-driven mechanisms. At the same time, although this study had a simple analysis of regional differences, it did not provide an in-depth analysis of regional-scale environmental regulation. As city clusters and metropolitan areas are important spatial carriers of regional development competition, the green development of manufacturing industries in different types of city clusters and metropolitan areas could be studied horizontally and comparatively in the future to enrich the theoretical framework of green development of manufacturing industries.

(3) The theoretical framework of this study needed to be strengthened to explore and verify the path. The tendency of green manufacturing development in Chinese cities with balanced development and the mismatch between the green manufacturing efficiency pattern and the economic scale pattern suggest that the sufficient condition to promote green development in manufacturing is not high economic factor input. In the empirical research, the green efficiency of the manufacturing industry is restrained by economic development, but the foundation of economic development determines the superstructure of development, and economic development is the key to realizing the sustainable development of cities. Therefore, it is obvious that the green development of the manufacturing industry contains complex mechanisms. On the one hand, the transmission effect mechanism of environmental regulation affecting green development in manufacturing and its path dependence can be further examined in the future by constructing interaction terms or mediating effects. On the other hand, subdividing manufacturing industries to explore the differential characteristics could allow examination of the power source of green development of manufacturing industries in more detail.

## Figures and Tables

**Figure 1 ijerph-19-11970-f001:**
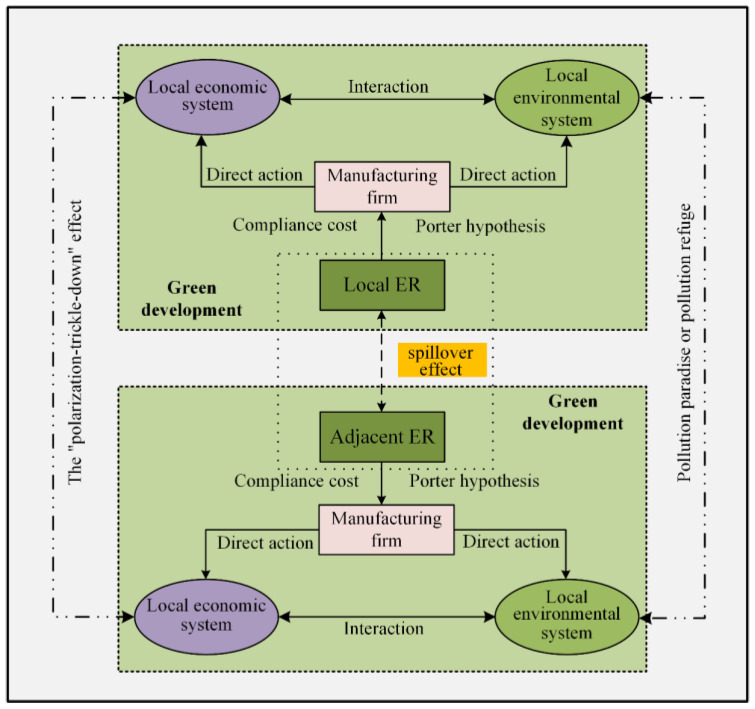
Theoretical framework.

**Figure 2 ijerph-19-11970-f002:**
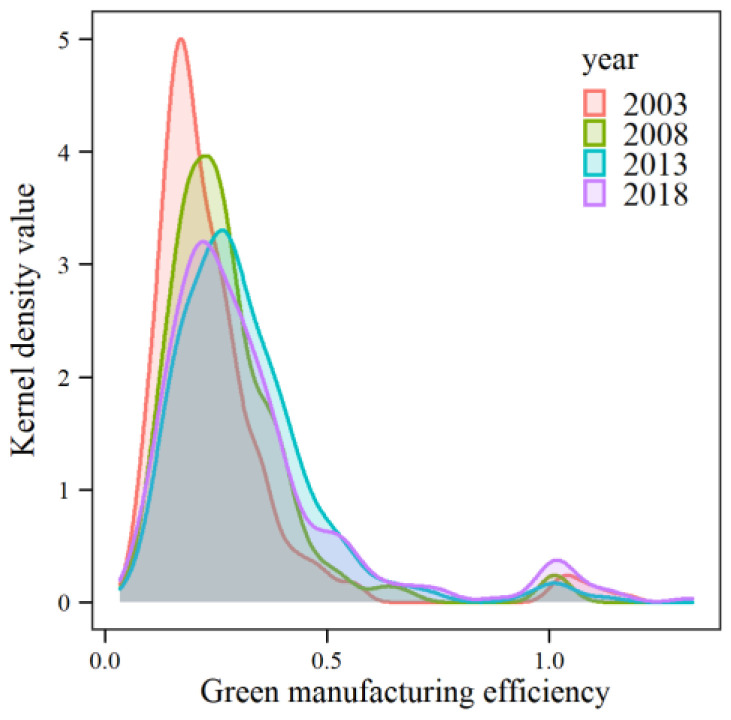
Kernel density of green manufacturing efficiency in China, 2003 to 2018.

**Figure 3 ijerph-19-11970-f003:**
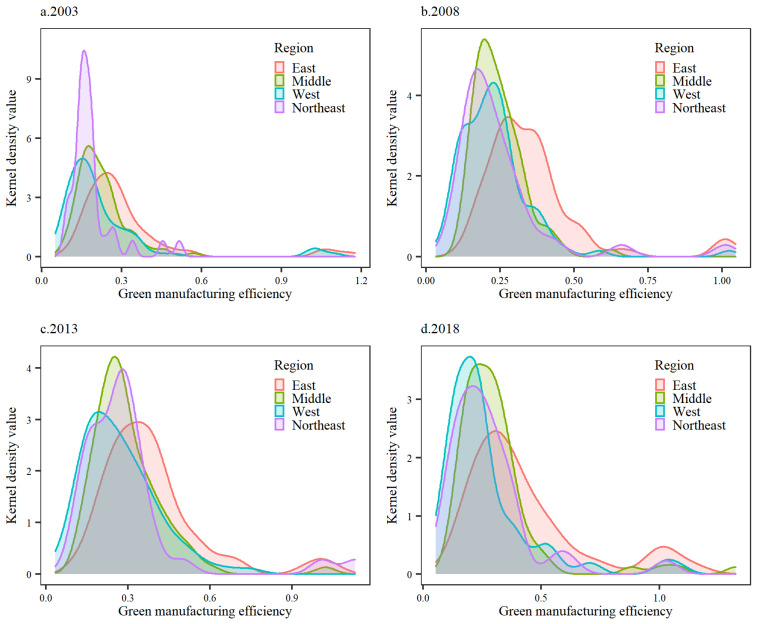
Kernel Density of Green Manufacturing Efficiency in China by Region, 2003 to 2018.

**Figure 4 ijerph-19-11970-f004:**
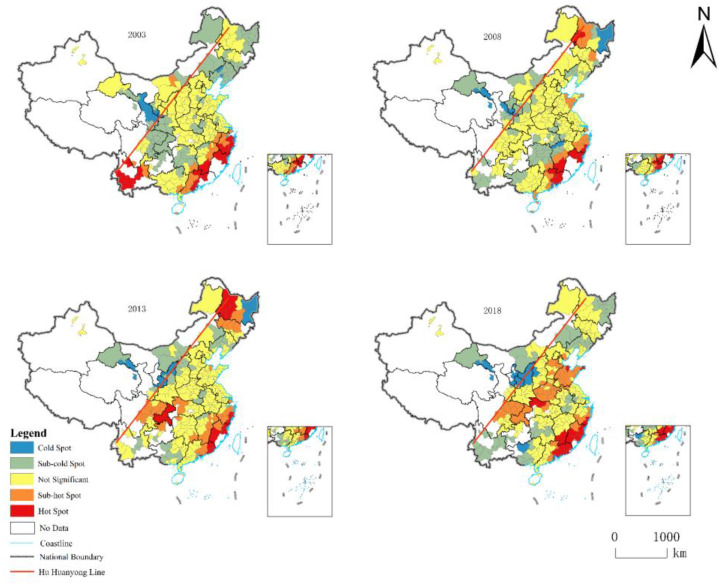
Analysis of urban cold hotspots in China, 2003 to 2018.

**Figure 5 ijerph-19-11970-f005:**
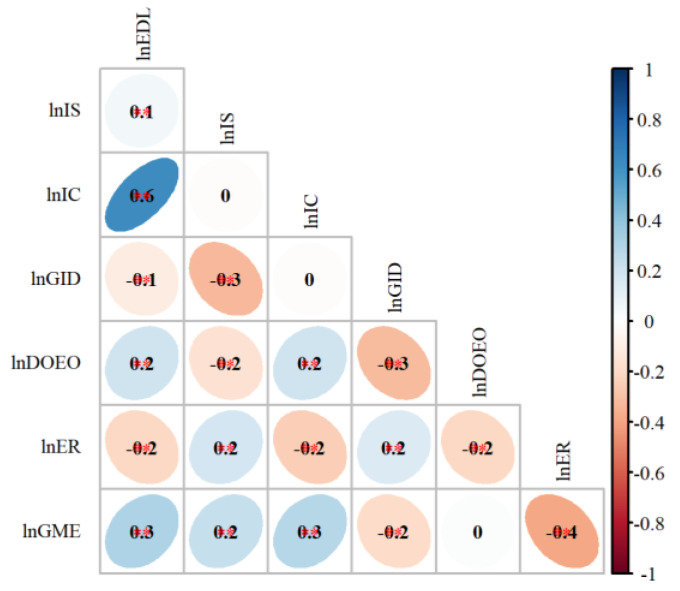
Plot of variable correlation coefficient test. Note: ** and * represent those whose significance levels are 0.01 and 0.05 respectively.

**Figure 6 ijerph-19-11970-f006:**
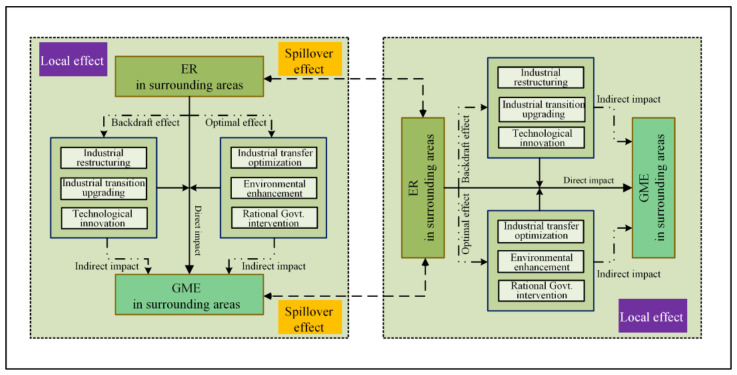
Theoretical Framework.

**Table 1 ijerph-19-11970-t001:** Descriptive statistics table.

Variable	Obs	Mean	Std. Dev.	Min	Max
*LnGME*	4544	−1.4061	0.5150	−3.7213	0.2797
*LnER*	4544	0.8346	0.6069	−2.7949	4.3102
*LnIS*	4544	10.1645	0.8360	4.5951	13.0557
*LnEDL*	4544	11.8795	1.9130	1.6094	16.9217
*LnIC*	4544	−0.2361	0.4443	−2.3591	1.4713
*LnGID*	4544	14.0877	1.1087	10.4058	18.2405
*LnDOEO*	4544	9.4171	2.2215	0.0000	14.9413

**Table 2 ijerph-19-11970-t002:** The collinearity diagnosis results.

Variable	VIF	1/VIF
*LnER*	1.17	0.8541
*LnIS*	1.20	0.8305
*LnEDL*	1.69	0.5919
*LnIC*	1.71	0.5851
*LnGID*	1.19	0.8385

**Table 3 ijerph-19-11970-t003:** Model test results.

Variables	$W_{d}$		$W_{e}$		$W_{ed}$	
	Coefficient	*p*	Coefficient	*p*	Coefficient	*p*
*Moran’s I*	41.163	0	21.885	0	33.385	0
*LM-error*	992.849	0	474.483	0	1021.81	0
*LM-lag*	1552.48	0	432.727	0	561.028	0
*LR-lag*	12.44	0.0293	10.91	0.0533	19.88	0.0013
*LR-error*	12.5	0.0285	17.4	0.0038	20.59	0.001
*Hausman*	−135.36	/	36.93	0	−114.15	/

**Table 4 ijerph-19-11970-t004:** Estimation results of the SDM.

Variables	$W_{d}$	$W_{e}$	$W_{ed}$
*LnER*	−0.0448 ***	−0.0427 ***	−0.0415 ***
	(−12.46)	(−12.15)	(−11.61)
*LnIS*	0.345 ***	0.339 ***	0.339 ***
	(19.30)	(18.81)	(18.88)
*LnEDI*	−0.0654 ***	−0.0601 ***	−0.0775 ***
	(−4.13)	(−3.81)	(−4.89)
*LnIC*	−0.00144	−0.0062	−0.00147
	(−0.19)	(−0.83)	(−0.20)
*LnGID*	0.0275	0.0574 **	0.0211
	(1.13)	(2.22)	(0.87)
*Wx * LnER*	0.0877 ***	0.0078	0.0125
	(2.89)	(1.34)	(0.44)
*Wx * LnIS*	0.271	0.0685 **	0.269 *
	(1.60)	(2.31)	(1.93)
*Wx * LnEDI*	−0.00660	−0.0392	0.186
	(−0.05)	(−1.53)	(1.49)
*Wx * LnIC*	−0.0378	−0.000917	−0.0796
	(−0.89)	(−0.08)	(−1.61)
*Wx * LnGID*	0.147	−0.0727 *	0.129
	(0.75)	(−1.84)	(0.77)
ρ	0.318 ***	0.112 ***	0.0136
	(3.07)	(5.75)	(0.12)
*R^2^*	0.0199	0.0326	0.106
*N*	4544	4544	4544

Note: The value of T is in brackets; ***, ** and * represent those whose significance levels are 0.01, 0.05 and 0.1 respectively.

**Table 5 ijerph-19-11970-t005:** Parametric decomposition of spatial spillover effect.

Decomposition Category	Variables	$W_{d}$	$W_{e}$	$W_{ed}$
Direct effect	*LnER*	−0.0444 ***	−0.0425 ***	−0.0414 ***
		(−12.07)	(−11.82)	(−11.28)
	*LnIS*	0.348 ***	0.343 ***	0.342 ***
		(20.38)	(20.02)	(19.80)
	*LnEDI*	−0.0661 ***	−0.0619 ***	−0.0781 ***
		(−4.35)	(−4.10)	(−5.11)
	*LnIC*	−0.00166	−0.00636	−0.00157
		(−0.23)	(−0.90)	(−0.22)
	*LnGID*	0.0280	0.0557 **	0.0211
		(1.17)	(2.21)	(0.88)
Indirect effect	*LnER*	0.111 **	0.00370	0.0130
		(2.45)	(0.59)	(0.46)
	*LnIS*	0.556 **	0.115 ***	0.274 **
		(2.36)	(4.02)	(2.06)
	*LnIC*	−0.0513	−0.000606	−0.0766
		(−0.83)	(−0.05)	(−1.54)
	*LnEDI*	−0.0371	−0.0502 *	0.193
		(−0.18)	(−1.77)	(1.47)
	*LnGID*	0.243	−0.0712 *	0.146
		(0.83)	(−1.68)	(0.83)
Total effect	*LnER*	0.0664	−0.0388 ***	−0.0284
		(1.48)	(−5.56)	(−1.01)
	*LnIS*	0.903***	0.459 ***	0.615 ***
		(3.87)	(14.99)	(4.73)
	*LnEDI*	−0.103	−0.112 ***	0.115
		(−0.51)	(−3.65)	(0.88)
	*LnIC*	−0.0530	−0.00696	−0.0782 *
		(−0.90)	(−0.63)	(−1.65)
	*LnGID*	0.271	−0.0154	0.167
		(0.95)	(−0.39)	(0.99)

Note: The value of T is in brackets; ***, ** and * represent those whose significance levels are 0.01, 0.05 and 0.1 respectively.

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
