# Peer review of "Spatiotemporal Differences and Spatial Spillovers of China’s Green Manufacturing under Environmental Regulation"

_ijerph, 2022, doi:10.3390/ijerph191911970_

Round 1

Reviewer 1 Report

After reading the reviewed article, I can say the following:

1. The main issue of the article concerns the analysis of the impact of environmental regulations on the efficiency of ecological production and the spatial effect in China. Exploratory spatial analysis was used to characterize the spatiotemporal differentiation of urban green manufacturing efficiency in the years 2003-2018. The Durbin spatial model was selected for the research.
2. The article is well written, both in terms of content and editing. Due to the complex problems presented in this article, it is quite difficult for the reders.
3. The aauthors of the article, however, are aware of many shortcomings of this article and the need to conduct further resarch in this area.
4. Literature is reach and fully complements the authors' deliberations.
5. Notes organizing your work were introduced in track changes mode

The reviewer suggests publishing this article, after cosidering the commments, in the International Journal of Environmental Research and Public Health.

Reviewer 2 Report

By furnishing substantial experiments, this paper discussed the spatiotemporal differences and spatial spillovers of China’s green manufacturing under environmental regulation which is a meaningful topic. The authors conducted experiments from multi-aspects to illustrate their results. Some points that may be helpful for readers better understand this research include:

1) Several methods (SBM-DEA model, kernel density estimation, hot spot analysis, space panel durbin model) were used to present and analysis the data, what the logical relationship between them? This relates to the logic of the research, and may need to be illustrated.

2) The data are not described in Part 3.

3) The results of correlation and collinearity of the control variables are presented, but their respective discussion are missing. The correlation and collinearity are necessary to be discussed because it relates to the reliability of the model.

Some details:

1) line 20: …indicating that manufacturing (unnecessary?) green manufacturing is not traditionally dominated…

2) line 162-163: …manufacturing industry data are used instead of manufacturing industry data in this paper…  A are used instead of B, A and B are usually defferent.

3) the indent issues of explaining the formulae, “where…” may be not indented, like line 188, 199, ……

4) Necessary references need to be included. Line 225, Tobler’s work of the first law of geography; line 455, Since Lesage pointed…; line 348, Porter proposed that…

5) X in equation (6) are not explained. GDP?

6) Figure 2 appears at the beginning of Part 4.1.1, it may be better to appear after its respect text.

7) Some paragraphs may be too long, and it is easier for readers to reader shorter paragraph. So it is kind of authors to partition those long paragraphs, like lines 251-284, lines 288-316, etc.

8) line 343, …is used as the explanatory variable. I suppose the GME is the explained variable.

9) line 353, …regula-tion…

10) lines 403-404, The results show that the spatial correlations of different matrix models pass the test and that they all pass the LM significance test.

I understand that authors used these tests to test the significance of spatial autocorrelation parameter of spatial lagged term (table 2), and the results show significant p values which indicates that the null hypothesis (no spatial autocorrelation) should be rejected. So why authors states “pass” the test?

11) line 650, …findings were as follows. Were->are, present tense for results.

12) line 651, a space is needed between (1) and China’s…

Round 2

Reviewer 2 Report

4.2.2 “Model selection test and model reconstruction” is an important section of this research, so some issues may need to be clarified:

1) the model selection tests are selecting what? This relates to:

       - control variables, or spatial lag parts (including wy, wx, or lagged error)

     Or

-      The W matrix (which one, W_e or W_ed, is finally adopted in the model)?

It is helpful if authors can make this clear. (i.e., what is being tested?)

2) about expressing the test results:

  It may be more proper to express like this: the null hypothesis is XXX, the XXX test shows a significant result so we rejected the null hypothesis…

Some details:

Line 436. Hausman test results showed negative statistics under the ? and spatial weight matrices.

Line 443. Some Unnecessary spaces in the line.

Line 446. A space is needed before alpha_1GME (or it looks like a dot product with lnGME_i,t)

Line 494. Table 3 -> table 4
